# Lead-Free Multiferroic Barium-Calcium Zirconate-Titanate & Doped Nickel Ferrite Composites

Inna V. Lisnevskaya * , Inga A. Aleksandrova and Artem N. Savinov

Faculty of Chemistry, Southern Federal University, 344090 Rostov-on-Don, Russia
* Correspondence: liv@sfedu.ru

**Abstract:** Magnetoelectric lead-free composite ceramic based on the piezoelecrtic $Ba_{0.85}Ca_{0.15}Ti_{0.9}Zr_{0.1}O_3$ (BCZT) and magnetic $NiCo_{0.02}Cu_{0.02}Mn_{0.1}Fe_{1.8}O_{4-d}$ (NCCMF) has been obtained by the solid state method using preliminarily synthesized by the solid-state method precursors. X-ray diffraction measurements, microstructural, magnetic, dielectric, piezoelectric and magnetoelectric studies have been carried out. Impurity phases were not contained in the composites, and there were no signs of interfacial interaction even at the doping level. Ceramics has a high electrical resistivity at direct current (~$10^9$ $\Omega \cdot cm$) and, over the entire range of x studied, exhibits a combination of magnetic and piezoelectric parameters, which vary over a wide range and clearly depend on the composites composition. The maximum magnetoelectric coupling coefficient $\Delta E / \Delta H \approx 90$ mV/(cm·Oe) at a frequency of 1 kHz has been observed for specimens with x = 60–70%.

**Keywords:** multiferroics; magnetoelectric composite; lead-free ceramics; barium-calcium titanate-zirconate

## 1. Introduction

The progressive evolution of electronics, computing and telecommunications, cosmonautics, medicine and other industries continuously actualizes the problem of designing new materials with a certain set of predetermined magnetic, electrical, optical, thermophysical, piezoelectric, superconducting and other characteristics. They are usually called "functional", "smart" or "intellectual". These materials are able to independently, i.e., without human intervention or control actions, respond to changing external conditions, which open up wide and diverse prospects for their application.

Multiferroics, which belong to functional materials, are an extensive class of single-phase materials (chemical compounds and solid solutions based on them, [1–4]) and heterogeneous magnetostrictive-piezoelectric systems (magnetoelectric (ME) composites, [5–8]), which have the ability to mutually convert an electric and magnetic fields energy. Multiferroics can be used to control electrical polarization when a magnetic field is applied and magnetization when an electric field is applied, which is achieved at the material level, bypassing complex multi-node electrical circuits, resulting in reliability and efficiency. Multiferroics are excellent candidates for electrically written and magnetically read memory technologies, ambient sensors of magnetic field, energy harvesting devices, electrical field tunable devices and current/voltage converters and so on. Thereby, the searching and designing of new high-efficient multiferroic materials is a problem in materials science.

In recent years, the attention of particular researchers has been attracted by lead-free multiferroic compositions, both single-phase (for example, bismuth ferrite and solid solutions based on it and related systems, [9–12]) and heterogeneous [13–16] (particularly, ME composites based on barium-calcium titanate-zirconate ($Ba_{0.85}Ca_{0.15}(Zr_{0.1}Ti_{0.9})O_3$, BCZT)), due to limitation on the Pb-containing materials use in electronic equipment and other areas because of lead toxicity.

Barium-calcium zirconate-titanate ($Ba_{0.85}Ca_{0.15}(Zr_{0.1}Ti_{0.9})O_3$, BCZT) is one of the few lead-free piezoelectric materials that are not substandard in piezoelectric properties to

the best members of lead-containing piezoceramics. In 2009, W. Liu and X. Ren [17] were the first to report on a new lead-free system characterized by a tetragonal-rhombohedral morphotropic interfacial boundary near which the compositions have an extremely high piezoelectric coefficient $d_{33} > 500$ pC/N. Thus, 50 mol.% BZT–50 mol.% BCT piezoceramic (or, what is the same, $Ba_{0.85}Ca_{0.15}(Zr_{0.1}Ti_{0.9})O_3$, BCZT) shows the following characteristics: spontaneous polarization $P_s \approx 20$ μC/cm², remanent polarization $P_r \approx 15$ μC/cm², coercive force $E_c \approx 0.168$ V/mm, piezoelectric coefficient $d_{33} \approx 500$–600 pC/N, and relative permittivity $\varepsilon/\varepsilon_0 \approx 3060$. Its significant disadvantage is, first of all, the low Curie temperature (~110 °C), which sufficiently limits its potential use temperature range, as well as the high sintering temperature (~1400 °C). Nevertheless, there are a lot of studies on the search for dopant and on the variation of synthesis and sintering conditions aimed at improving the technological parameters of obtaining properties of $Ba_{0.85}Ca_{0.15}(Zr_{0.1}Ti_{0.9})O_3$ piezoceramics. Data on BCZT and solid solutions based on it were summarized in the article [18].

BCZT is of great interest to ME composites developers. In recent years, there have been many publications reporting that this material has been used as a piezoelectric phase in two-phase magnetostrictive-piezoelectric systems. Thus, Praveen et al. [19] obtained an ME composite ceramic $0.7(Ba_{0.85}Ca_{0.15})(Zr_{0.1}Ti_{0.9})O_3$–$0.3CoFe_2O_4$ from piezoelectric and ferrite powders synthesized by low-temperature methods, and a comprehensive study of its properties was carried out. The maximum transverse and longitudinal ME coefficients values in a field of 1.65 kOe are 7 mV/cm·Oe and 3.6 mV/cm·Oe, respectively; at the resonance frequency (365 kHz), the composite showed a maximum ME coefficient value of 102 mV/cm·Oe. Naveed-Ul-Haq [20] studied the dielectric properties of 85 wt.% $[0.5Ba(Zr_{0.2}Ti_{0.8})O_3 - 0.5(Ba_{0.7}Ca_{0.3})TiO_3] - 15$ wt.% $[CoFe_2O_4]$ composite ceramics obtained from piezoelectric and ferrite powders synthesized by the solid state method. Mixed and layered $[xCoFe_2O_4 (CFO) - (1 - x)(Ba_{0.85}Ca_{0.15}) (Zr_{0.1}Ti_{0.9})O_3 (BCZT)]$ composites were described in articles [21,22]. The original auto-combustion technique was used to synthesize CFO nanoparticles, which were added to the BCZT precursor gel, then the composite powders were calcined at 800 °C and sintered at 1300 °C to obtain particulate composites. Laminate heterostructures were obtained by sandwiching piezoelectric and ferrite plates by silver epoxy bonding. All composites demonstrated the ME effect, while for mixed composites the maximum ME coupling coefficient at the resonance frequency was 104–161 mV/(cm·Oe) for the 40CFO-60BCZT specimen; however the highest ME coefficients were achieved by laminate composites, ~803 mV/(cm·Oe) for a trilayer and ~615 mV/(cm·Oe) for a bilayer heterostructure at the electromechanical resonance frequency. $(1 - x)(Ba_{0.85}Ca_{0.15}Zr_{0.1}Ti_{0.9}O_3) - x (Ni_{0.7}Zn_{0.3}Fe_2O_4)$ composites with high ME coupling efficiency were obtained by the traditional solid state method, where the ME coefficients were studied in both longitudinal and transverse modes and amounted to ~14.5 and ~13 mV/(E·cm), respectively, for a composite with a phase ratio of 50:50 at off-resonance frequency of 1 kHz, described in [23]. A significant increase in the ME coupling efficiency up to ~5.5 V/(cm·Oe) was observed at the electromechanical resonance frequency, which is a valuable result. However, the ME coupling efficiency significantly depends on the ferrite magnetostrictive properties. Thus, the transverse magnetoelectric coupling coefficient of the $Ba_{0.85}Ca_{0.15}Zr_{0.1}Ti_{0.9}O_3$-$MgFe_2O_4$ composites obtained by the solid state method in [24] were found to be low, 7.97 mV/(cm·Oe) at a frequency of 10 kHz. Thereby, there are a lot of experimental data indicating the effectiveness of using BCZT as a piezoelectric phase in two-phase multiferroic ceramics.

This manuscript examines a comprehensive study of the structure and properties of $(100 - x)$ wt.% $Ba_{0.85}Ca_{0.15}Ti_{0.9}Zr_{0.1}O_3$ (BCZT) + x wt.% $NiCo_{0.02}Cu_{0.02}Mn_{0.1}Fe_{1.8}O_{4-d}$, (NCCMF) ME composites, synthesized by solid state method. The ferrite composition was chosen based on the fact that it was previously [15,25] used in the Pb-containing composite ceramics produce and proved its high efficiency. Solid solutions based on nickel ferrite are widely used in electroacoustic transducers. They have high magnetostriction constants, high mechanical quality factor, and corrosion resistance. Concurrently, it is known [26] that small additions of cobalt ferrite significantly increase the dynamic magnetostrictive param-

eters of nickel ferrite due to the compensation of magnetocrystalline anisotropy, practically without affecting the static magnetostriction. The iron deficiency in comparison with the stoichiometric composition and the substitution of some iron ions with cobalt helps the production of nickel ferrite with a specific electrical resistance of more than $10^{10}$ Ohm·cm. Several iron ions can be replaced with manganese to increase the electrical resistance. It also seems expedient to dope ferrite with small additions of copper oxide, which reduces the sintering temperature and improves the MS characteristics of nickel ferrite.

## 2. Experimental

Nickel ferrite $NiCo_{0.02}Cu_{0.02}Mn_{0.1}Fe_{1.8}O_{4-d}$ (NCCMF) and barium-calcium zirconate-titanate $Ba_{0.85}Ca_{0.15}Ti_{0.9}Zr_{0.1}O_3$ (BCZT) were synthesized by solid state method according to the reactions:

$$NiO + 0.02CoO + 0.02CuO + 0.1MnO_2 + 0.9Fe_2O_3 = NiCo_{0.02}Cu_{0.02}Mn_{0.1}Fe_{1.8}O_{4-d}$$

$$0.85BaCO_3 + 0.15CaCO_3 + 0.9TiO_2 + 0.1ZrO_2 = Ba_{0.85}Ca_{0.15}Ti_{0.9}Zr_{0.1}O_3 + 0.85CO_2$$

The required amounts of oxides and carbonates were thoroughly ground, briquetted and then calcined for 8 h, NCCMF at 900–1000 °C, BCZT at 1200–1350 °C, with intermediate remixing. The ferrite powder was crushed and sieved through a 0.074 mm sieve before composites making.

The required mass ratios of piezoelectric and ferrite powders were mixed, pressed and sintered at 1200 °C to obtain x wt.% BCZT + (100 − x) wt.% NCCMF composite ceramics with x = 20–90 in increments of 10%. Pellets with a thickness of 1 mm were made from the obtained specimens, then electrodes were applied by burning a silver-containing paste at 500 °C and polarized in a pulsed mode in a $CCl_4$ at room temperature for 2–3 min with applied field of 3–4 kV/mm.

The control of the synthesis completeness and the study of the phase formation were carried out by X-ray diffraction (XRD) measurements by the powder method on an ARL-X'tra diffractometer with $CuK_{\alpha}$ radiation, in the range of $2\Theta = 20–70$ with a 4°/min scanning rate. The surface and bulk compositions were investigated by means of SEM (electron microscope VEGA II LMU with energy dispersive microanalysis system INCA EN-ERGY,450/XT equipped with an X-Act ADD detector). Electrical resistivity was measured by an E6-13A teraohmmeter under direct current. The specimens density was calculated from the mass and the volume measured from the linear dimensions. The composites dielectric properties were measured using a «Censurka-M» apparatus at a 1 kHz frequency. The temperature and frequency dependences of the dielectric constant on alternating current were studied according to the Sawyer-Tower circuit using an E7-20 immitance meter. The piezoelectric coefficient $d_{33}$ was measured by a quasi-static method on $d_{33}$-meter. The magnetoelectric (ME) properties were measured at a 1 kHz frequency with a special setup; its scheme is described in [27], with overlapping vectors of alternating (19.2 Oe) and permanent (0–2 kOe) magnetic fields and the electric polarization of ceramic specimens.

## 3. Results and Discussion

Figure 1 shows the X-ray diffraction pattern of (100 − x) wt.% BCZT + x wt.% NCCMF composite ceramics. It can be seen that the specimens consist of two phases, perovskite and spinel; no impurity phases were found. The diffraction peaks positions of the composites matches with those for pure phases. It may indicate the absence of interfacial interaction even at the doping level. An expected intensity increase of the spinel peaks and intensity decrease of the perovskite peaks are observed with ferrite content increasing.

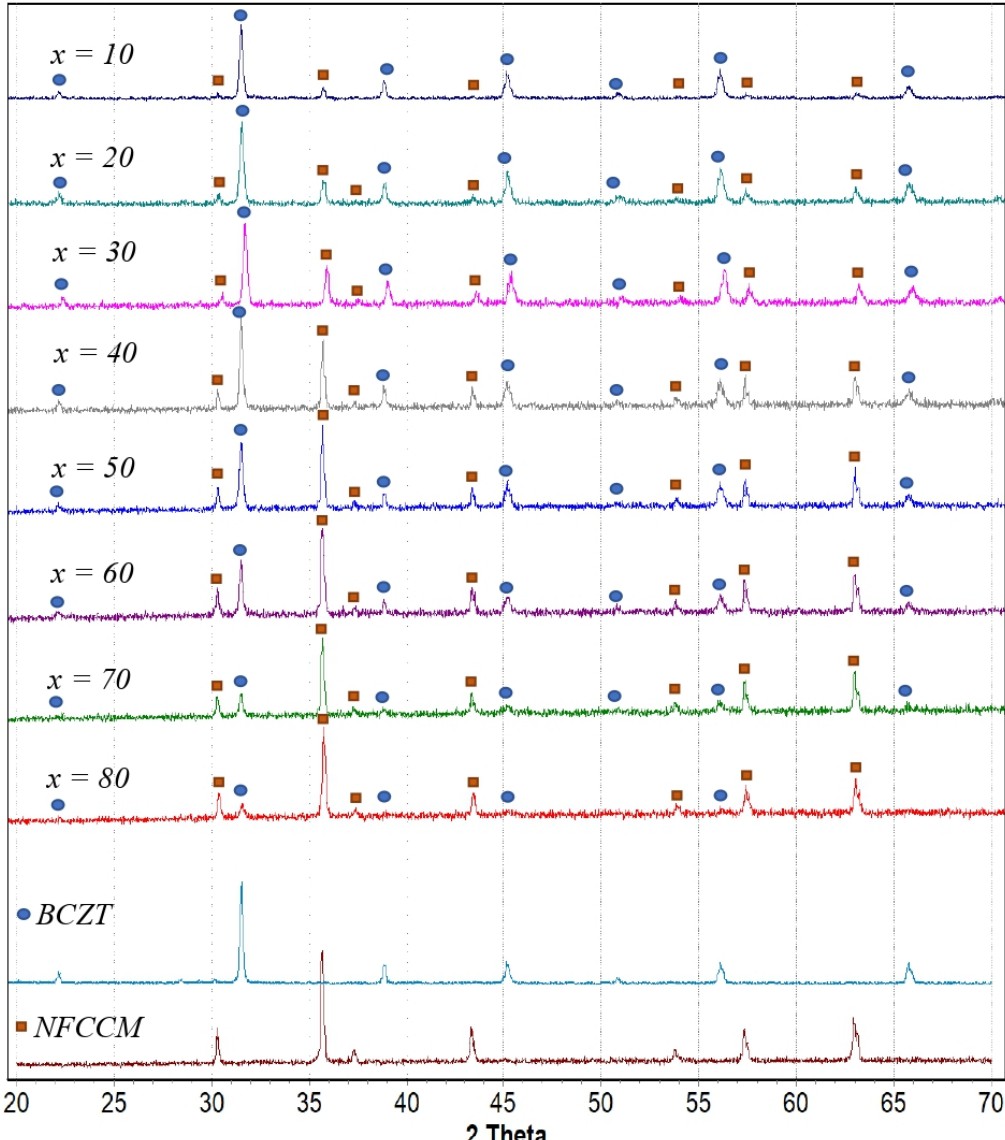

**Figure 1.** Powder XRD patterns of (100 − x) wt.% BCZT + x wt.% NCCMF composites and pure components.

SEM images of (100 − x) wt.% BCZT + x wt.% NCCMF composite ceramics are presented in Figures 2 and 3. Foremost, the regular differences in the phases volume content depending on x (Figure 2a–h) and the clearly defined particles contours of each of the phases at their contact points are clearly visible, which may indirectly indicate the absence of interpenetration and chemical interaction of the phases in the composites sintering process. It can be seen that the particle size of the piezoelectric and magnetostrictive phases is significantly different. The BCZT phase particles have an irregular shape due to piezoelectric precursor prehistory: as mentioned, in order to reduce interfacial interaction, the piezomaterial was pre-sintered at 1400 °C as a dense ceramic and then grounded in a mortar, so many BCZT particles have an irregular, chipped form. The size of some BCZT particles reaches 20 μm or more, but there is also a large size dispersion, many of the particles are smaller by an order of magnitude or more, ~1–2 μm (Figure 3a). NCCMF phase particles are smaller and more uniform in size, <1 μm. With an increase in the ferrite content in the composites, the predominant 0–3 connectivity type of piezoelectric-ferrite is more and more clearly achieved, due to strong differences in the particle sizes of the piezoelectric and magnetostrictive phases, which is shown in the example in Figure 3b–e for composites with high x at lower magnification.

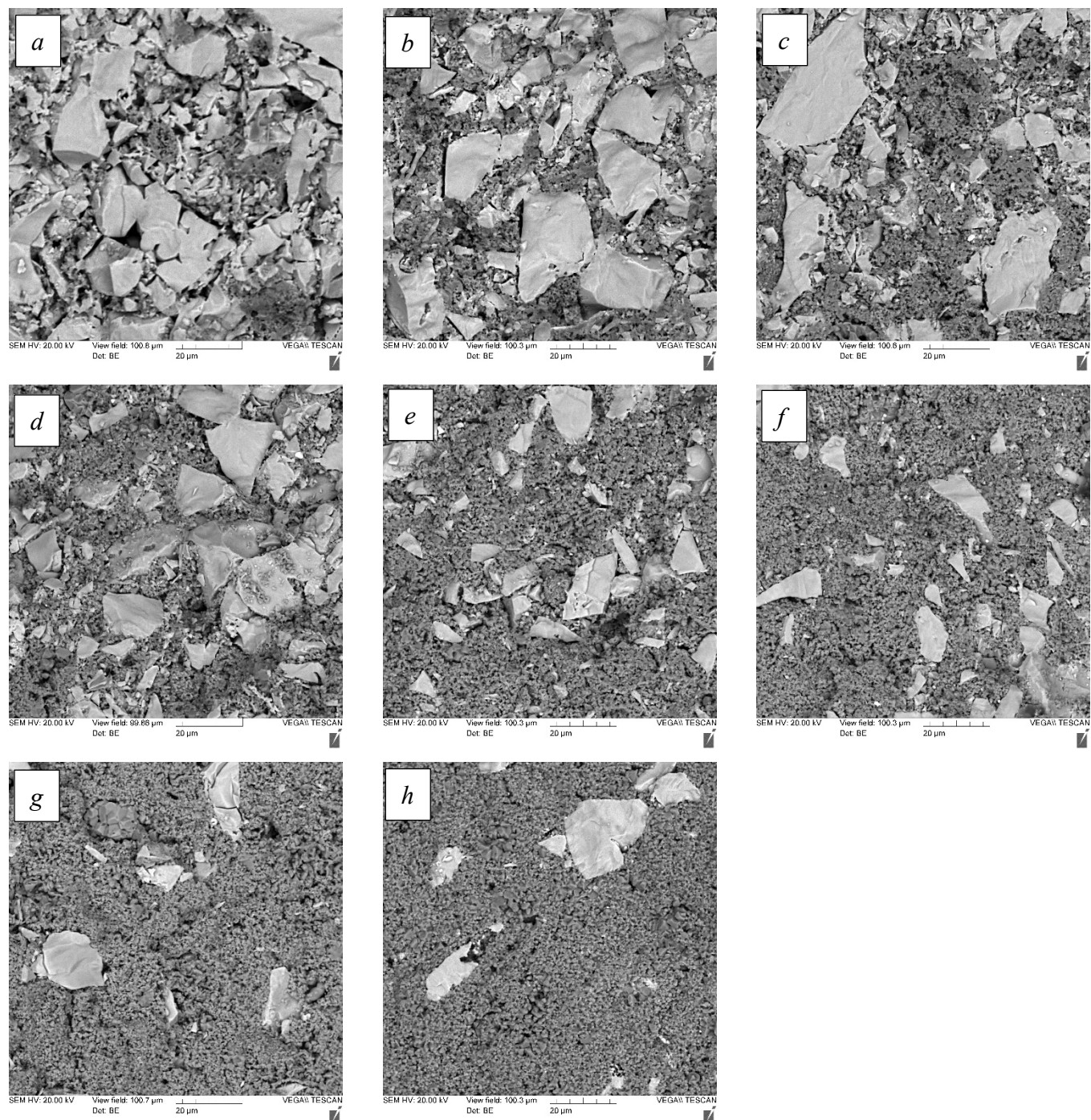

**Figure 2.** SEM images of (100 − x) wt.% BCZT + x wt.% NCCMF composite ceramics, (**a**) x = 20, (**b**) x = 30, (**c**) x = 40, (**d**) x = 50, (**e**) x = 60, (**f**) x = 70, (**g**) x = 80, (**h**) x = 90.

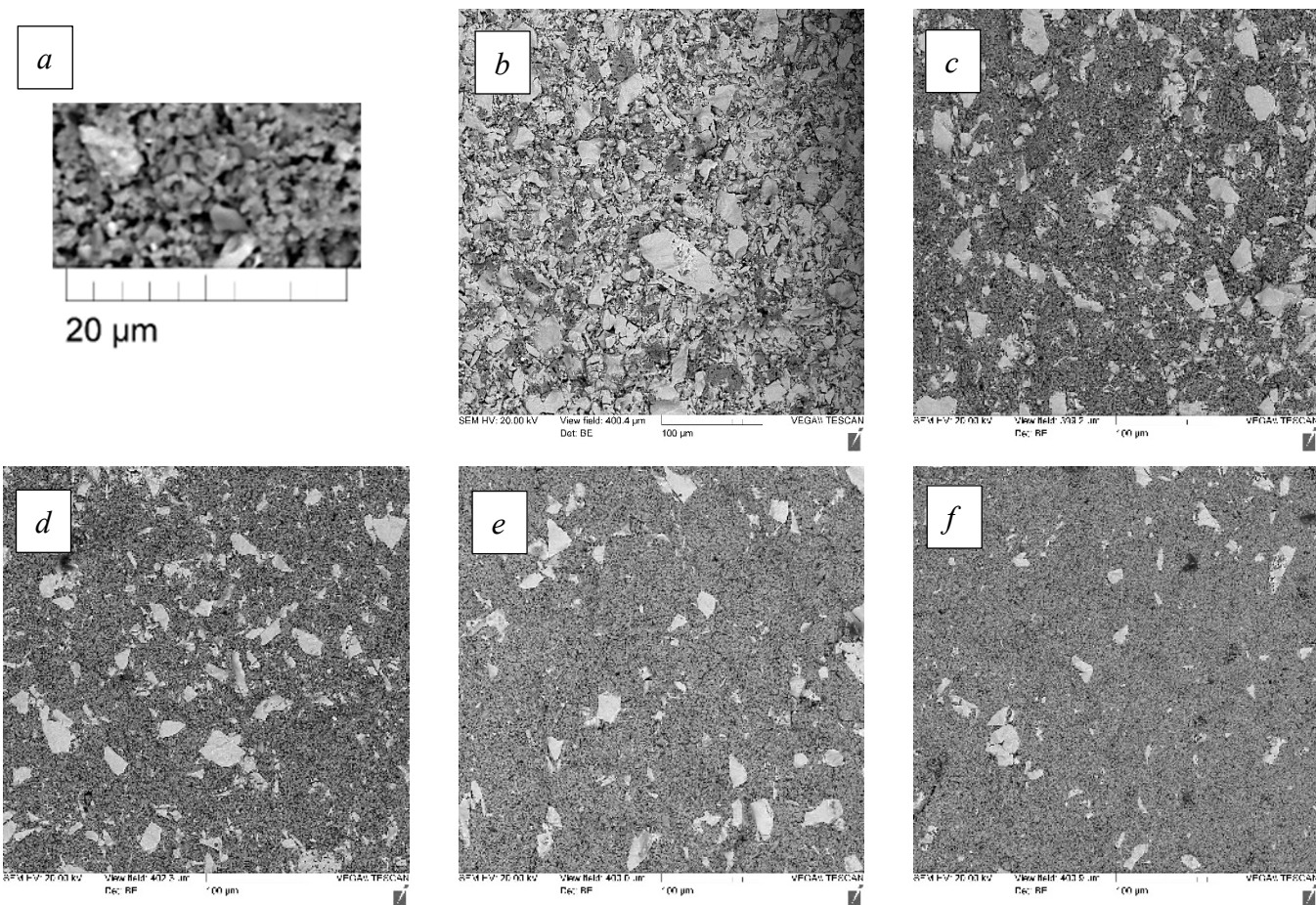

**Figure 3.** SEM images of (100 − x) wt.% BCZT + x wt.% NCCMF composite ceramics, (**a**) ferrite grain morphology, (**b**) microstructure of the composite x = 20 with a predominant 3-0 connectivity type, (**c**–**f**) illustration of the predominant 0-3 connectivity type in composites with x = 60–90.

Figure 4 shows the magnetic hysteresis loops of (100 − x) wt.% BCZT + x wt.% NCCMF, and the values of the coercive force, saturation magnetization and remanent magnetization are listed in Table 1. With an increase in the magnetostrictive component content composites, a predictable saturation magnetization increase is observed (from ~3.8 to ~31 emu/g at x = 10 and 80 wt.%, respectively) with a coercive force decrease (from ~46 to ~20 kOe). It is known that the coercive force Hc is a structurally sensitive property of materials. Thus, the Hc values for the same material in the single-crystal and polycrystalline states are often several orders of magnitude smaller. Especially in composites, the coercive force of which should increase with increasing dilution of the magnetically active phase with the nonmagnetic, which is observed in (100 − x) wt.% BCZT + x wt.% NCCMF composite ceramics. This is due to the motion of domain walls, which is hampered at grain boundaries in single-phase materials and even more so in composites. Remanent magnetization Mr is another structurally sensitive parameter of materials. As can be seen from Table 1, it changes in a peculiar way for (100 − x) wt.% BCZT + x wt.% NCCMF composites, passing through a wide maximum in the range x = 30–70 (the point x = 30 does not follow this pattern, which can be counted as an artifact). Presumably, this is due to a change in the predominant type of connectivity. At x < 20, the microstructure of the composites is a single grain of ferrite in a piezoelectric matrix (3-0 connectivity type, Figure 3b); at x > 70, on the contrary,—piezoelectric grains in the ferrite matrix (0-3 connectivity type, Figure 3e,f); at intermediate values of x, a mixed type of connectivity is observed, including the 3-3 connectivity. At small values of x, the remanent magnetization Mr of the composites is low due to the low content of the magnetic component in the specimens, and with an increase

in x, the MR increases accordingly. In composites with a mixed type of connectivity, ferrite grains are in contact both with each other and with piezoelectric grains, and in composites with a 0-3 connectivity, they are predominantly in contact with each other. The decrease in Mr of composites with a higher content of the magnetostrictive component is observed due to the easier demagnetization process at the grain boundaries of ferrite-ferrite than at the ferrite-piezoelectric.

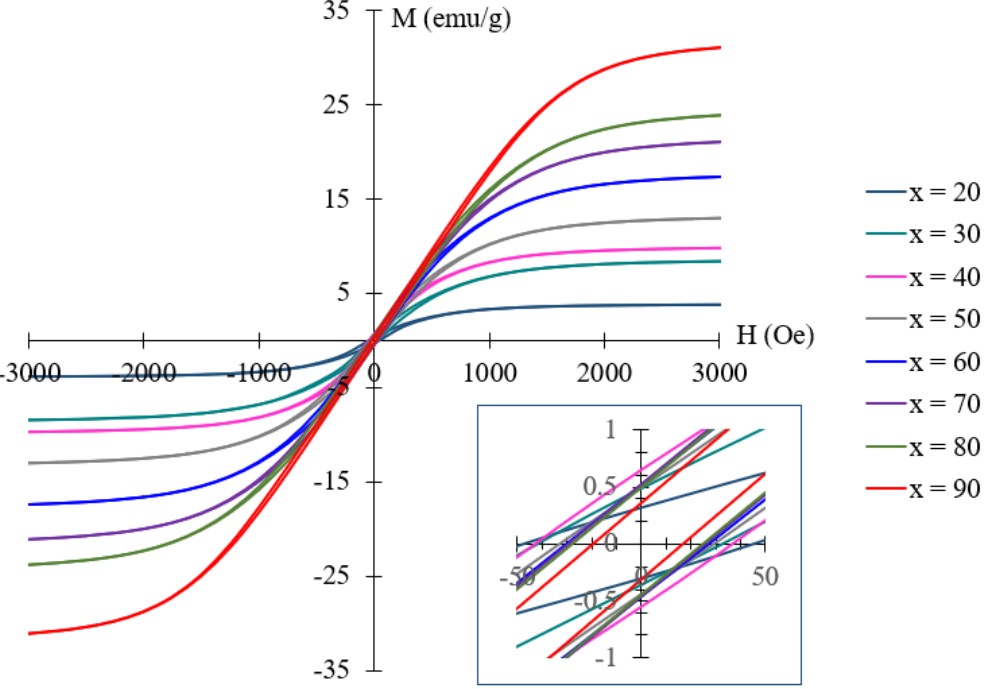

**Figure 4.** Magnetic hysteresis loops of (100 − x) wt.% BCZT + x wt.% NCCMF composite ceramics.

**Table 1.** Properties of ME composite ceramics (100 x) wt.% BCZT + x wt.% NCCMF.

| x | $M_S$, emu/g | $M_R$, emu/g | $H_C$, Oe | log R | $\varepsilon/\varepsilon_0$ | tan δ | $d_{33}$, pC/N | $g_{33}$, mV/(m·N) | ΔE/ΔH, mV/(cm·Oe) |
|---|---|---|---|---|---|---|---|---|---|
| 20 | 3.8 | 0.32 | 46 | 9.2 | 709 | 0.09 | 37.2 | 6.1 | 22.5 |
| 30 | 8.5 | 0.43 | 43 | 9.5 | 365 | 0.11 | 33.6 | 10.4 | 49.9 |
| 40 | 9.7 | 0.65 | 42 | 9.8 | 324 | 0.12 | 22 | 10.8 | 52.6 |
| 50 | 13 | 0.5 | 34 | 9.1 | 155 | 0.33 | 18.6 | 13.5 | 81.1 |
| 60 | 17.4 | 0.52 | 31 | 8.8 | 113 | 0.55 | 16.3 | 16.3 | 89.7 |
| 70 | 21 | 0.52 | 30 | 7.9 | 71 | 0.89 | 9 | 14.3 | 90.6 |
| 80 | 23.8 | 0.5 | 28 | 8.5 | 46 | 0.77 | 5.3 | 13 | 70.7 |
| 90 | 31 | 0.36 | 20 | 7.7 | 44 | 0.99 | 2.2 | 5.5 | 50 |

Further, Figure 5 shows the concentration dependences of density ρ, dielectric constant $\varepsilon/\varepsilon_0$, electrical resistivity logarithm log R and dielectric loss tangent tan δ of (100 − x) wt.% BCZT + x wt.% NCCMF composite ceramics. The values of ρ and $\varepsilon/\varepsilon_0$ (Figure 5a,b) and tan δ increases (Figure 5c) with an increase in the ferrite content. This is because the intrinsic density and dielectric constant of the piezomaterial is higher, and the loss tangent is lower than that of ferrite. At the same time, the specimen electrical resistance at direct current changes rather insignificantly (Figure 5d).

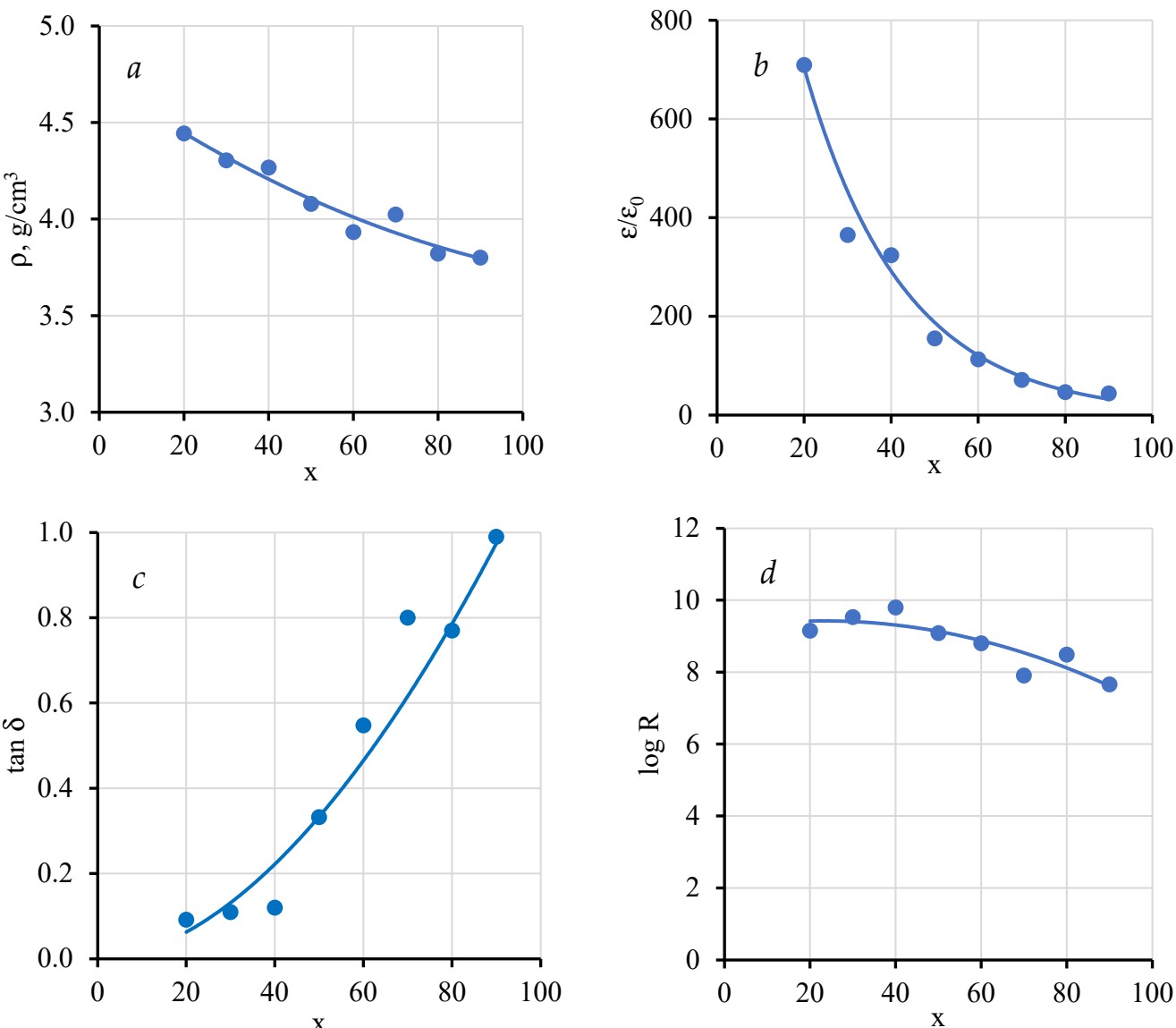

**Figure 5.** Concentration dependences of density (**a**), dielectric constant (**b**), electrical resistivity logarithm (**c**) and dielectric loss tangent (**d**) of $(100 - x)$ wt.% BCZT + x wt.% NCCMF composite ceramics.

Figure 6 shows the dielectric constant $\varepsilon/\varepsilon_0$ temperature dependences of $(100 - x)$ wt.% BCZT + x wt.% NCCMF composite ceramics, measured in an alternating electric field at a frequency of 1 MHz (the data of specimens with a high magnetostrictive component content in the frequency range up to 10 kHz are incorrect due to the significant influence of the specimens electrical conductivity on the measuring signal magnitude). It can be seen that with an increase in the ferrite content, the permittivity value at the maximum decreases naturally and quite sharply. Herewith, the ferroelectric Curie point corresponds to the temperature (which is close to the Curie point of pure BCZT) and slightly increases with x increasing. It is possibly due to a change in the composites mechanical characteristics in the presence of a ferrite component. The magnetostrictive phase compresses the piezoelectric grains, preventing the phase transition to the paraelectric state due to greater elastic stiffness, which leads to an increase in the phase transition temperature. In the high frequency range (50–1000 kHz) in the studied temperature interval (20–200 °C), the composites dielectric constant slightly varies, and frequency dispersion is absent. Dielectric measurements at temperatures above 200–250 °C are incorrect due to the increasing of ceramics electrical conductivity.

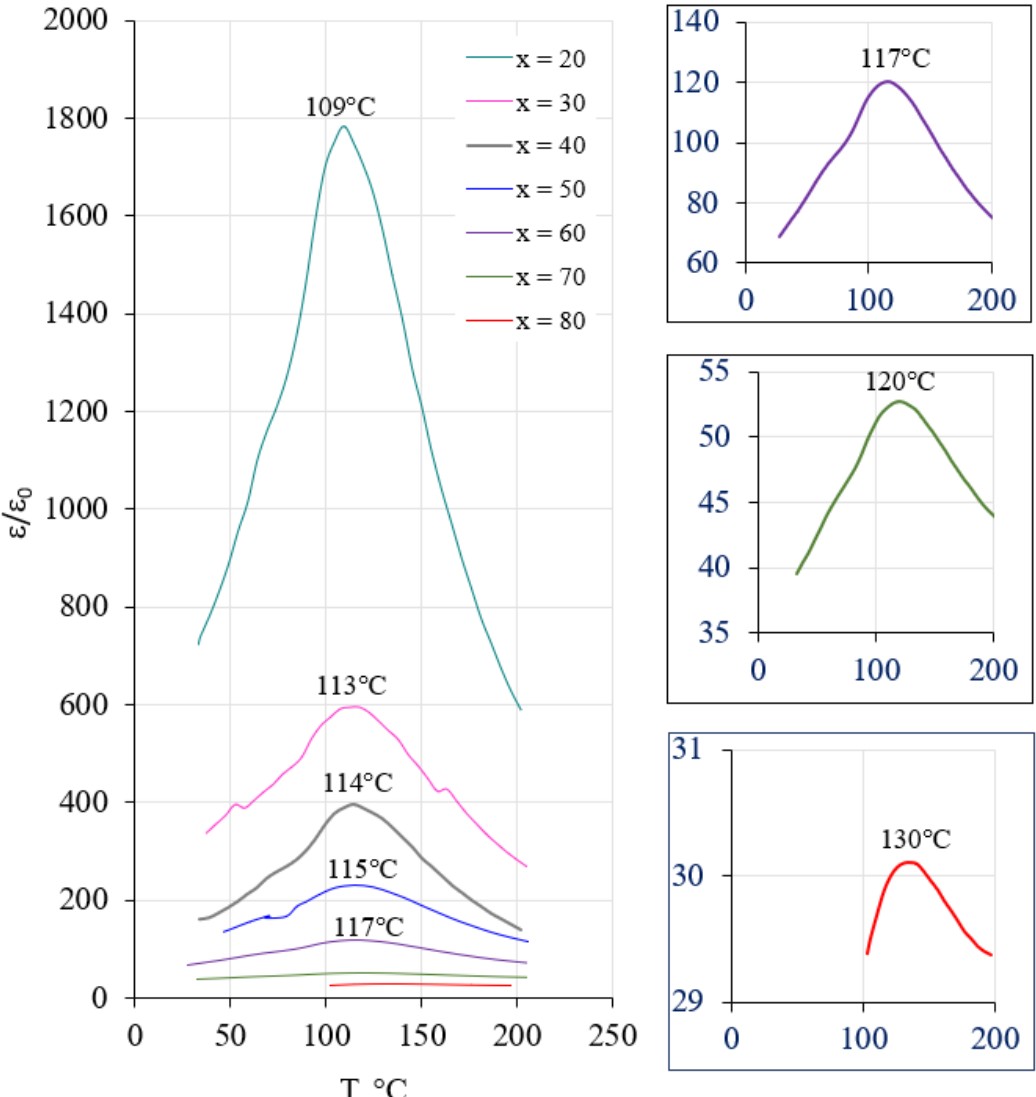

**Figure 6.** The dielectric constant temperature dependences of $(100 - x)$ wt.% BCZT + x wt.% NCCMF composite ceramics at a frequency of 1 MHz.

The piezoelectric coefficient naturally decreases (Figure 7a) with an increase in the ferrite content, which is associated with the dilution of the active piezophase with inactive ferrite. The piezoelectric voltage constant $g_{33}$ (Figure 7b) passes through a maximum as a result of the two parameters influencing the superposition—the dielectric constant and the piezoelectric voltage constant ($g_{33} = d_{33}/\varepsilon_{33}^{T}$). Consequently, the ME coupling coefficient $\Delta E/\Delta H$ (Figure 7c) also passes through a maximum in the 60–70 wt.% NCCMF composition range, thus, a regular correlation between the $g_{33}$ and $\Delta E/\Delta H$ parameters of particulate composites based on $Ba_{0.85}Ca_{0.15}Ti_{0.9}Zr_{0.1}O_3$ и $NiCo_{0.02}Cu_{0.02}Mn_{0.1}Fe_{1.8}O_{4-d}$ is revealed.

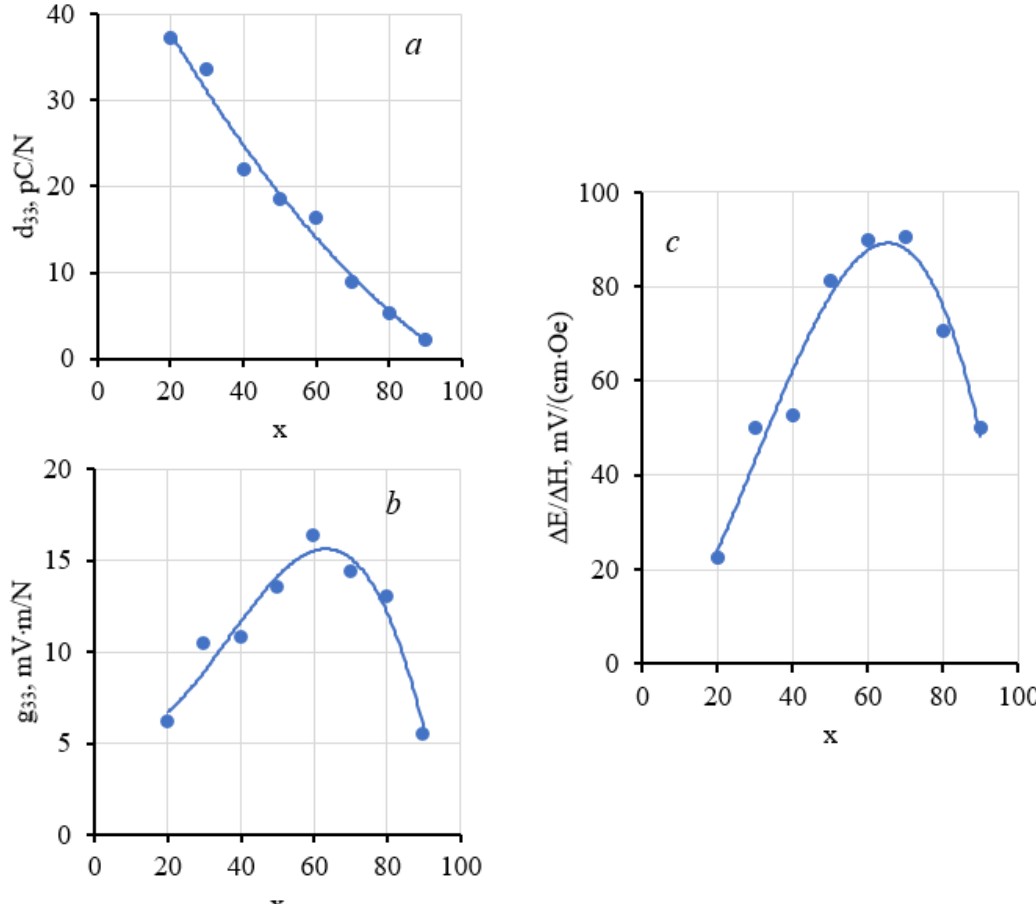

**Figure 7.** Concentration dependences of the piezoelectric coefficient (**a**), piezoelectric voltage constant (**b**), and ME coupling coefficient (**c**) of (100 − x) wt.% BCZT + x wt.% NCCMF composites.

The obtained lead-free ME ceramics is not inferior in terms of piezo parameters and ME coupling efficiency to materials of the lead zirconate titanate group previously obtained by the solid state method [13]. Moreover, the obtained lead-free ME ceramics even slightly exceed the lead-containing analogues in terms of piezoelectric parameters.

Finally, it should be mentioned that the synthesis and sintering temperature of BCZT and composite ceramics based on it can be significantly reduced by using low-temperature methods. There are experimental data on composites (100 − x) wt.% BCZT + x wt.% NCCMF, where BCZT was obtained from citrate gel at a temperature of 600–700 °C, and composite ceramics was obtained at a temperature of 1000–1050 °C. However, the piezoelectric properties of composite materials were extremely low, and the ME coupling coefficients were zero. Thus, it is not so difficult to reduce the temperature to obtain ME ceramics, but the properties of BCZT are very structurally sensitive. This is similar to $BaTiO_3$, which, as we know, almost completely loses its piezoactivity in a finely dispersed state [28].

## 4. Conclusions

Lead-free ME composite ceramic (100 − x) wt.% $Ba_{0.85}Ca_{0.15}Ti_{0.9}Zr_{0.1}O_3$ (BCZT) + x wt.% $NiCo_{0.02}Cu_{0.02}Mn_{0.1}Fe_{1.8}O_{4-d}$, (NCCMF), without any impurity phases, as well as signs of interfacial interaction at the doping level, was obtained by the solid state method. Ceramics has a high electrical resistivity at direct current (~$10^9$ Ω·cm) and demonstrates a combination of magnetic and piezoelectric properties in the entire x range, which vary over a wide range and naturally depend on the composition of the composites. The maximum ME coupling coefficient ΔE/ΔH ≈ 90 mV/(cm·Oe) has been observed for specimens with x = 60–70 wt.%. The saturation magnetization and remanent magnetization of these specimens are, respectively, ~22 and ~0.5 emu/g, the coercive force is ~30 kOe,

the dielectric constant at room temperature at a frequency of 1 kHz and 1 MHz is ~90 and ~70, respectively, for x = 60 and ~55 and ~40 for x = 70. The dielectric loss tangents at the indicated frequencies are 0.4–0.5 and ~0.03, respectively. The piezoelectric coefficient for specimens with x = 60 and 70 is 16 and 9 pC/N; the piezoelectric voltage constant is 16.3 and 14.3 mV·m/N, respectively.

**Author Contributions:** Conceptualization, I.V.L.; methodology, I.V.L.; validation, I.V.L., A.N.S. and I.A.A.; investigation, I.A.A. and A.N.S.; writing—review and editing, I.V.L. and I.A.A.; visualization, I.V.L. and I.A.A.; All authors have read and agreed to the published version of the manuscript.

**Funding:** This research received no external funding.

**Institutional Review Board Statement:** The study did not require ethical approval.

**Informed Consent Statement:** Informed consent was obtained from all subjects involved in the study.

**Data Availability Statement:** All the data supporting reported results is given in the article. Additional datasets are not suggested.

**Acknowledgments:** We are grateful to Yu. V. Rusalev, The Smart Materials Research Institute, SFedU for the magnetic measurements and to Yu. V. Popov, Center for Research of Mineral Raw Materials and the State of the Environment, Institute of Earth Sciences, SFedU for the SEM analysis.

**Conflicts of Interest:** The authors declare no conflict of interest.

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
