# Peer review of "Lead-Free Multiferroic Barium-Calcium Zirconate-Titanate & Doped Nickel Ferrite Composites"

_jcs, doi:10.3390/jcs7010002_

Round 1

Reviewer 1 Report

The work is focused on investigates the lead-free magnetoelectric composites (100-х) wt.% Ba0.85Ca0.15Ti0.9Zr0.1O3 (BCZT) + х wt.% NiCo0.02Cu0.02Mn0.1Fe1.8O4-d, (NCCMF), x = 10-80, which were synthesized by the solid-state method. This work is relevant and focused on investigating the physical properties of these structures. In the work the results of the X-ray diffraction measurements, microstructural, magnetic, dielectric, piezoelectric and magnetoelectric properties have been carried out. The abstract is concise and clear. The introduction and experimental sections are written clearly and very well organized. It is quite easy to grasp the motivation of the work. The author accurately explains how the data was obtained. The experiment can be reproduced by other researchers.

Discussion is strong. The authors good show the meaning of the results.

The references good reflect the main publications in this field.

In summary, the manuscript is scientifically sound, and one looks good. It is sufficiently novel and interesting research, and one is to warrant publication with minor revision.

1) On the P.4 Fig 2 need to add the caption NCCMF.

2) The caption j must be moved from page 5 to page 6.

Author Response

We are greatly appreciate to the respected reviewer for his attention to the manuscript and hope that the article can be published after the recommended corrections have been made. Our answers:

1) On the P.4 Fig 2 need to add the caption NCCMF.

Thanks to the reviewer for this comment. We have made the necessary changes to the figure.

2) The caption j must be moved from page 5 to page 6.

According to the recommendation of another reviewer, figure 2 was divided into two, now they are Figures 2 and 3. The notation has changed accordingly. We have checked that they are not misaligned with respect to the figures.

We are also ready to answer any additional questions.

Reviewer 2 Report

In the present work, the authors have successfully synthesized lead-free composite ceramic and comprehensively studied its structure and properties. Piezoelectric materials have already found wide applications in various fields, but the most widely used materials for commercial applications contain Pb (such as PZT), which has adverse effects on human health and the environment and this has motivated the active development of lead-free alternatives. Therefore the present article is relevant and interesting, and the main conclusions of the work were confirmed experimentally. My comments are listed below.

1) The quality of the presentation can be improved. So, in Figure 1 dotted lines should be removed, as well as add the caption "NCCMF" for the square label. Additionally, it would be better to shift the labels of the most intense peaks (for example, place them to the right of the peaks themselves), so that they do not overlap the XRD patterns; In Figure 2, the caption "j" is outside the corresponding image, and the captions "k", "l", and "m" protrude beyond the images, which should be corrected.

2) The article needs proofreading to correct some misprints. For example, in the introduction one of the ME coefficient values is indicated as ~5.5 V/(E*cm) instead of ~5.5 V/(Oe*cm), and so on.

Author Response

We are greatly appreciate to the respected reviewer for his attention to the manuscript and hope that the article can be published after the recommended corrections have been made. Our answers:

 1) The quality of the presentation can be improved. So, in Figure 1 dotted lines should be removed, as well as add the caption "NCCMF" for the square label. Additionally, it would be better to shift the labels of the most intense peaks (for example, place them to the right of the peaks themselves), so that they do not overlap the XRD patterns; In Figure 2, the caption "j" is outside the corresponding image, and the captions "k", "l", and "m" protrude beyond the images, which should be corrected.

We are grateful to the respected reviewer for the remark. On Fig. 1 we have added the missing inscription. The labels above the peaks were indeed shifted, we corrected this and carefully positioned them above the peaks. According to the recommendation of another reviewer, Figure 2 was divided into two, now they are Figures 2 and 3. The notation has changed accordingly. We have checked that they are not misaligned with respect to the figures.

2) The article needs proofreading to correct some misprints. For example, in the introduction one of the ME coefficient values is indicated as ~5.5 V/(E*cm) instead of ~5.5 V/(Oe*cm), and so on.

Thanks, this misprint has been corrected.

We are also ready to answer any additional questions.

Reviewer 3 Report

The work presented in this manuscript covers a relevant topic for the field of functional materials, and showcases the advantages of designing composite structures for multiferroics. Although the motivation is clear in the introduction section, it would be a nice hook for the readers if a succinct answer to the "so what?" is added to the abstract. The majority of the experiments are well conducted and the results are properly presented and discussed. A few questions and suggestions for the revvised version:

1. In Introduction, when explaining why this particular ferrite composition was chosen, "based on the fact that it was previously [15, 25] used in the Pb-containing composite ceramics produce and proved its high efficiency." the papers were author's own. It is alright to motivate the materials choice based on your own previous work, but the authors must give a better and clearer justification for the choice of materials and cite a broader audience encompassing other researchers. It is suggested that authors add a checklist to help the community choose different materials to make such composites?

2. Early in the introduction, it is mentioned that a high sintering temperature is a concern for the manufacturability of these materials. However, the sintering temperatures used for the components here are still 900-1350 degC. So, it must be explained in the paper on how this work solves that issue. Further, the authors must add in the conclusions section, an outlook for how to proceed with such materials, to meet that temperature requirement? Perhaps, authors can add potential materials that others could work with.

3. If the data presented in Figures 1, 3, and 5 can be represented with the same color scheme for all x ranges, it would be easier for the readers to grasp the data. For example, all plots with x=10 could be blue, x =20 cyan, and so forth.

4. Line 2 of page 5, the abbreviation for nickel ferrite is misspelt as NFCCM instead of NCCMF.

5. It is recommended that Figure 2 SEM images be split into 2 separate figures. One for compositions only (a through h), other for specifics of ferrite grain morphology and connectivity type (i through m).

6. It is mentioned that predictable saturation magnetization increase is observed across the composition. If predictable, then how come saturation magnetization for x=20% is larger than x=30%? Same concern arises for the coercive force. Further, why is remnant magnetization similar for all composites? All of these must be discussed in the manuscript, and the sentence about the results being predictable, must be changed for accuracy of the data.

7. It must be mentioned in the Methods section as to how the density shown in Figure 4a was calculated.

8. Ensure the y-axis must read logR (not lgR), and same in the text, where there is a typo of lgR and tgD, instead of logR and tanD, respectively.

9. Figure 5 must include the phase transtion temperature for x = 80.

Also, there are several instances throughout the manuscript that are difficult to understand and are in need of extensive English language editing. Some examples have been added below:

1. "Such materials are usually called "functional", as well as "smart", "intellectual", "intellectual"" Seems irrelevant, also intellectual repeated.

2. Sentence is difficult to understand "The control of the ferrite and piezoelectric material synthesis completeness and the study of the composites phase formation were carried out by X-ray"

3. Must simplify this sentence "The diffraction peaks positions of the compo-sites matches with those for pure phases, which can be indicative of the interfacial inter-action absence, including at the doping level. An.."

4. Kindly simplify this sentence for ease of reading "Herewith, the ferroelectric Curie point corresponds to the temperature (which is close to the Curie point of pure BCZT) and slightly increases with x increasing, which possibly due to a change in the composites mechanical characteristics in the presence of a ferrite component: the magnetostrictive phase compresses the piezoelectric grains, preventing the phase transition to the paraelectric state due to greater elastic stiffness, which leads to an increase in the phase transition temperature."

5. Need English styling to make this concluding sentence more commanding. "ME ceramic obtained as a result of this study is not inferior in terms of piezo parameters and ME coupling efficiency, and even slightly superior lead-containing analogs in terms of piezo parameters."

6. The following long sentence "The maximum ME coupling coefficient Е/H  90 mV/(cmOe) have been observed for specimens with x = 60-70 wt. %, for which the saturation magnetization and remanent magnetization are respectively 22 and 0.5 emu/g, the coercive force is 30 kOe, the dielectric constant at room temperature at a frequency of 1 kHz and 1 MHz is ~90 and ~70 , respectively, for x = 60 and ~55 and ~40 for x = 70, the dielectric loss tangent at the indicated frequencies is 0.4–0.5 and ~0.03, respectively, the piezoelectric coefficient for specimens with x = 60 and 70 is 16 and 9 pC/N, the piezoelectric voltage constant is 16.3 and 14.3 mV⋅m /N, respec-tively." must be broken down into multiple sentences. Rather, it is strongly recommended that the all parameters of the compositions must be symmarized in a table, and the best composition be highlighted.

Author Response

We are greatly appreciate to the respected reviewer for his attention to the manuscript and hope that the article can be published after the recommended corrections have been made. Our answers:

  1. In Introduction, when explaining why this particular ferrite composition was chosen, "based on the fact that it was previously [15, 25] used in the Pb-containing composite ceramics produce and proved its high efficiency." the papers were author's own. It is alright to motivate the materials choice based on your own previous work, but the authors must give a better and clearer justification for the choice of materials and cite a broader audience encompassing other researchers. It is suggested that authors add a checklist to help the community choose different materials to make such composites?

We are greatly appreciate the reviewer for this suggestion. The following explanation and the reference [26] have been added to the manuscript:

Solid solutions based on nickel ferrite are widely used in electroacoustic transducers. They have high magnetostriction constants, high mechanical quality factor, and corrosion resistance. Concurrently, it is known [26] that small additions of cobalt ferrite significantly increase the dynamic magnetostrictive parameters of nickel ferrite due to the compensation of magnetocrystalline anisotropy, practically without affecting the static magnetostriction. The iron deficiency in comparison with the stoichiometric composition and the substitution of some iron ions with cobalt helps to the production of nickel ferrite with a specific electrical resistance of more than 1010 Ohm•cm. Several iron ions can be replaced with manganese to increase the electrical resistance. It also seems expedient to dope ferrite with small additions of copper oxide, which reduces the sintering temperature and improves the MS characteristics of nickel ferrite.

  1. Early in the introduction, it is mentioned that a high sintering temperature is a concern for the manufacturability of these materials. However, the sintering temperatures used for the components here are still 900-1350 degC. So, it must be explained in the paper on how this work solves that issue. Further, the authors must add in the conclusions section, an outlook for how to proceed with such materials, to meet that temperature requirement? Perhaps, authors can add potential materials that others could work with.

The synthesis and sintering temperature of BCZT and composite ceramics based on it can be significantly reduced by using low-temperature methods. There are experimental data on composites (100 x) wt.% BCZT + x wt.% NCCMF, where BCZT was obtained from citrate gel at a temperature of 700 C, and composite ceramics was obtained at a temperature of 1050 C. However, the piezoelectric properties of composite materials were extremely low, and the ME coupling coefficients were zero. Thus, it is not so difficult to reduce the temperature to obtain ME ceramics, but the properties of BCZT are very structurally sensitive. Similarly with BaTiO3, which, as we know, almost completely loses its piezoactivity in a finely dispersed state. Thanks to the reviewer for the discussion.

  1. If the data presented in Figures 1, 3, and 5 can be represented with the same color scheme for all x ranges, it would be easier for the readers to grasp the data. For example, all plots with x=10 could be blue, x =20 cyan, and so forth.

Thanks to the reviewer for this comment. We have made the necessary changes to these figures. Now a specific x value corresponds to one color in all the figures.

  1. Line 2 of page 5, the abbreviation for nickel ferrite is misspelt as NFCCM instead of NCCMF.

Thanks, this misspelt has been corrected.

  1. It is recommended that Figure 2 SEM images be split into 2 separate figures. One for compositions only (a through h), other for specifics of ferrite grain morphology and connectivity type (i through m).

Figure 2 have been splitted into 2 separate figures, now they are Figures 2 and 3. The numbering of all subsequent figures has also been corrected.

  1. It is mentioned that predictable saturation magnetization increase is observed across the composition. If predictable, then how come saturation magnetization for x=20% is larger than x=30%? Same concern arises for the coercive force. Further, why is remnant magnetization similar for all composites? All of these must be discussed in the manuscript, and the sentence about the results being predictable, must be changed for accuracy of the data.

We thank the reviewer for this remark. The experimental data have been carefully checked and the necessary changes were made. Also, according to the recommendation of a respected reviewer, Table 1 has been added, where all data on composites have been entered. The following notes were also included in the manuscript:

Figure 4 shows the magnetic hysteresis loops of (100‑x) wt.% BCZT + x wt.% NCCMF, and the values of the coercive force, saturation magnetization and remanent magnetization are listed in Table 1. With an increase in the magnetostrictive component content composites, a prospective increase in the saturation magnetization is observed (from ~3.8 to ~31 emu/g at x = 10 and 80 wt.%, respectively) with a decrease in the coercive force (from ~46 to ~20 kOe). It is known that the coercive force Hc is a structurally sensitive property of materials. Thus, the Hc values for the same material in the single-crystal and polycrystalline states are often several orders of magnitude smaller. Especially in composites, the coercive force of which should increase with increasing dilution of the magnetically active phase with the nonmagnetic one, which is observed in (100- x) wt.% BCZT + x wt.% NCCMF composite ceramics. This is due to the motion of domain walls, which is hampered at grain boundaries in single-phase materials and even more so in composites. Remanent magnetization Mr is another structurally sensitive parameter of materials. As can be seen from Table 1, it changes in a peculiar way for (100-x) wt.% BCZT + x wt.% NCCMF composites, passing through a wide maximum in the range x = 30-70 (the point x = 30 does not follow this pattern, which can be count as an artifact). Presumably, this is due to a change in the predominant type of connectivity. At x < 20, the microstructure of the composites is a single grain of ferrite in a piezoelectric matrix (3-0 connectivity type); at x > 70, on the contrary, - piezoelectric grains in the ferrite matrix (0-3 connectivity type); at intermediate values of x, a mixed type of connectivity is observed, including the 3-3 connectivity. At small values of x, the remanent magnetization Mr of the composites is low due to the low content of the magnetic component in the specimens, and with an increase in x, the MR increases accordingly. In composites with a mixed type of connectivity, ferrite grains are in contact both with each other and with piezoelectric grains, and in composites with a 0-3 connectivity, predominantly with each other. The decrease in Mr of composites with a higher content of the magnetostrictive component is observed due to the easier demagnetization process at the grain boundaries of ferrite-ferrite than at the ferrite-piezoelectric.

  1. It must be mentioned in the Methods section as to how the density shown in Figure 4a was calculated.

Information has been added on measuring the specimens density by weighing them and measuring volume by linear dimensions.

  1. Ensure the y-axis must read logR (not lgR), and same in the text, where there is a typo of lgR and tgD, instead of logR and tanD, respectively.

We are grateful to the respected reviewer. We carefully reviewed the manuscript and corrected all these misspelts.

  1. Figure 5 must include the phase transtion temperature for x = 80.

Thanks for the note, the phase transition temperature for x = 80 have been indicated.

Also, there are several instances throughout the manuscript that are difficult to understand and are in need of extensive English language editing. Some examples have been added below:

  1. "Such materials are usually called "functional", as well as "smart", "intellectual", "intellectual"" Seems irrelevant, also intellectual repeated.
  2. Sentence is difficult to understand "The control of the ferrite and piezoelectric material synthesis completeness and the study of the composites phase formation were carried out by X-ray"
  3. Must simplify this sentence "The diffraction peaks positions of the compo-sites matches with those for pure phases, which can be indicative of the interfacial inter-action absence, including at the doping level. An.."
  4. Kindly simplify this sentence for ease of reading "Herewith, the ferroelectric Curie point corresponds to the temperature (which is close to the Curie point of pure BCZT) and slightly increases with x increasing, which possibly due to a change in the composites mechanical characteristics in the presence of a ferrite component: the magnetostrictive phase compresses the piezoelectric grains, preventing the phase transition to the paraelectric state due to greater elastic stiffness, which leads to an increase in the phase transition temperature."
  5. Need English styling to make this concluding sentence more commanding. "ME ceramic obtained as a result of this study is not inferior in terms of piezo parameters and ME coupling efficiency, and even slightly superior lead-containing analogs in terms of piezo parameters."
  6. The following long sentence "The maximum ME coupling coefficient DЕ/DH » 90 mV/(cm×Oe) have been observed for specimens with x = 60-70 wt. %, for which the saturation magnetization and remanent magnetization are respectively ~22 and ~0.5 emu/g, the coercive force is ~30 kOe, the dielectric constant at room temperature at a frequency of 1 kHz and 1 MHz is ~90 and ~70 , respectively, for x = 60 and ~55 and ~40 for x = 70, the dielectric loss tangent at the indicated frequencies is 0.4–0.5 and ~0.03, respectively, the piezoelectric coefficient for specimens with x = 60 and 70 is 16 and 9 pC/N, the piezoelectric voltage constant is 16.3 and 14.3 mV⋅m /N, respec-tively." must be broken down into multiple sentences. Rather, it is strongly recommended that the all parameters of the compositions must be symmarized in a table, and the best composition be highlighted.

We are greatly appreciate to the respected reviewer for carefully reading our manuscript. In accordance with comments 1-6 concerning the English Language, the text has been corrected.

We are also ready to answer any additional questions.

Round 2

Reviewer 1 Report

The work is focused on investigates the lead-free magnetoelectric composites (100-х) wt.% Ba0.85Ca0.15Ti0.9Zr0.1O3 (BCZT) + х wt.% NiCo0.02Cu0.02Mn0.1Fe1.8O4-d, (NCCMF), x = 10-80, which were synthesized by the solid-state method. This work is relevant and focused on investigating the physical properties of these structures. In the work the results of the X-ray diffraction measurements, microstructural, magnetic, dielectric, piezoelectric and magnetoelectric properties have been carried out. The abstract is concise and clear. The introduction and experimental sections are written clearly and very well organized. It is quite easy to grasp the motivation of the work. The author accurately explains how the data was obtained. The experiment can be reproduced by other researchers.

Discussion is strong. The authors good show the meaning of the results.

The references good reflect the main publications in this field.

In summary, the manuscript is scientifically sound, and one looks good. It is sufficiently novel and interesting research, and one is to warrant publication without revision.

Author Response

We are grateful to you for revision of our manuscript and for the favourable decision about the possibility of its publication. 

Reviewer 2 Report

The changes made by the authors to the revised version of the manuscript have improved the quality of the report and the article can be accepted in its current form.

Author Response

(The authors gave the same response as above.)

Reviewer 3 Report

The authors have made several changes to the manuscript and added meaningful explanation of their results. A few minor suggestions for the final version:

1. Recommend adding some explanation similar to the text from the authors reply to the comments to the manuscript "The synthesis and sintering temperature of BCZT and composite ceramics based on it can be significantly reduced by using low-temperature methods. There are experimental data on composites (100 x) wt.% BCZT + x wt.% NCCMF, where BCZT was obtained from citrate gel at a temperature of 700 C, and composite ceramics was obtained at a temperature of 1050 C. However, the piezoelectric properties of composite materials were extremely low, and the ME coupling coefficients were zero. Thus, it is not so difficult to reduce the temperature to obtain ME ceramics, but the properties of BCZT are very structurally sensitive. Similarly with BaTiO3, which, as we know, almost completely loses its piezoactivity in a finely dispersed state." Adding this text to the introduction section with relevant references will be helpful to the readers. 

2. Although mentioned in the table 1, the data for x=90 is missing from the plots in Figures 5 and 7 and must be included in the final version.

3. Figure 6 is missing the dielectric constant temperate-dependence data for x=90. This must be explained in the text.

Author Response

We appreciate your attention for our manuscript. You helped to make our article better. Responses to your comments follow below. We are ready to answer all your questions.

  1. Recommend adding some explanation similar to the text from the authors reply to the comments to the manuscript "The synthesis and sintering temperature of BCZT and composite ceramics based on it can be significantly reduced by using low-temperature methods. There are experimental data on composites (100 x) wt.% BCZT + x wt.% NCCMF, where BCZT was obtained from citrate gel at a temperature of 700 C, and composite ceramics was obtained at a temperature of 1050 C. However, the piezoelectric properties of composite materials were extremely low, and the ME coupling coefficients were zero. Thus, it is not so difficult to reduce the temperature to obtain ME ceramics, but the properties of BCZT are very structurally sensitive. Similarly with BaTiO3, which, as we know, almost completely loses its piezoactivity in a finely dispersed state." Adding this text to the introduction section with relevant references will be helpful to the readers. 

We thank the distinguished referee for this recommendation. We have added this explanation to the manuscript in the "Results and discussions" section, as we refer to the obtained data, which have not been published yet. Reference [28] was also added describing the effect of barium titanate grains sizes on its piezoelectric properties.

  1. Although mentioned in the table 1, the data for x=90 is missing from the plots in Figures 5 and 7 and must be included in the final version.

Many thanks. We have included the x=90 point to all plots in Figures 5 and 7.

  1. Figure 6 is missing the dielectric constant temperate-dependence data for x=90. This must be explained in the text.

Unfortunately, the dielectric constant temperate-dependence data for x = 90 cannot be measured due to the increased electrical conductivity of the specimens.

Round 3

Reviewer 3 Report

Authors have made sufficient edits and changes to their manuscript based on the comments from all referees. However, the references 10-16 in the introduction section are self-citations and not yet been replaced by works of other researchers of the community. In my opinion, once some of these self-cited references are replaced by works of others, and only relevant authors papers are used as references, the manuscript can be accepted for publication.

"In recent years, particular researchers attention has been attracted by lead-free multiferroic compositions, both single-phase (for example, bismuth ferrite and solid solutions based on it and related systems, [9-12]) and heterogeneous [13-16]..."